# Novel Epigenetic Modifiers of Histones Presenting Potent Inhibitory Effects on Adenoid Cystic Carcinoma Stemness and Invasive Properties

**DOI:** 10.3390/ijms25031646

**Published:** 2024-01-29

**Authors:** Paulo S. S. Pina, Yeejin Jang, Carolina Emerick, João Figueira Scarini, Suzana C. O. M. Sousa, Cristiane H. Squarize, Rogerio M. Castilho

**Affiliations:** 1Laboratory of Epithelial Biology, Department of Periodontics and Oral Medicine, School of Dentistry, University of Michigan, Ann Arbor, MI 48109, USA; ps.souzapina@alumni.usp.br (P.S.S.P.); stellajang23@gmail.com (Y.J.); carolina.emerick@gmail.com (C.E.); scarinij@gmail.com (J.F.S.); csquariz@umich.edu (C.H.S.); 2Department of Stomatology, School of Dentistry, University of São Paulo, Sao Paulo 05508-270, Brazil; scmsouza@usp.br; 3Oral Diagnosis Department, Piracicaba School of Dentistry, State University of Campinas, Piracicaba 13414-903, Brazil; 4Rogel Cancer Center, University of Michigan, Ann Arbor, MI 48109, USA

**Keywords:** epigenetic, salivary gland cancer, HDAC, drug screening, EMT, cancer stem cell

## Abstract

Adenoid cystic carcinoma (ACC) is a rare neoplasm known for its indolent clinical course, risk of perineural invasion, and late onset of distant metastasis. Due to the scarcity of samples and the tumor’s rarity, progress in developing effective treatments has been historically limited. To tackle this issue, a high-throughput screening of epigenetic drugs was conducted to identify compounds capable of disrupting the invasive properties of the tumor and its cancer stem cells (CSCs). ACC cells were screened for changes in tumor viability, chromatin decondensation, Snail inhibition along tumor migration, and disruption of cancer stem cells. Seven compounds showed potential clinical interest, and further validation showed that Scriptaid emerged as a promising candidate for treating ACC invasion. Scriptaid demonstrated a favorable cellular toxicity index, effectively inhibited Snail expression, induced hyperacetylation of histone, reduced cell migration, and effectively disrupted tumorspheres. Additionally, LMK235 displayed encouraging results in four out of five validation assays, further highlighting its potential in combating tumor invasion in ACC. By targeting the invasive properties of the tumor and CSCs, Scriptaid and LMK235 hold promise as potential treatments for ACC, with the potential to improve patient outcomes and pave the way for further research in this critical area.

## 1. Introduction

Adenoid cystic carcinoma (ACC), a rare malignant neoplasm, generally impacts the minor oral salivary glands. The disease is known for its slow clinical progression, high likelihood of perineural invasion and local recurrence, marginal probability of lymph node metastasis, clear histopathological features, and resistance to conventional treatments. Late onset of distant metastasis is also a characteristic of ACC. Often, the disease remains undetected until it invades local nerves, which triggers symptoms and prompts patients to seek medical consultation [1,2,3,4].

Regardless of the primary lesion site, the view persists that the ACC responds poorly to available therapies, and there are no agreed-upon standard therapies for managing ACC. Most therapies include surgical removal of the lesion, followed by radiotherapy [5]. However, studies on therapies reliant on platinum-based compounds have shown less than satisfactory outcomes [3]. At present, numerous clinical trials for ACC are in progress, exploring the use of treatments such as Cetuximab, Paclitaxel, Bortezomib, and Sunitinib, among others. However, achieving a complete response to the disease remains limited [6].

Although most tumor cells can be destroyed with radiotherapy and chemotherapy, cancer stem cells (CSCs) are typically resistant, leading them to multiply and cause disease recurrence [7]. Therefore, targeting CSCs through histone acetylation, an epigenetic process involved in gene regulation, could be a viable strategy.

The epithelial–mesenchyme transition (EMT) significantly contributes to the invasion and growth of neoplastic cells from malignant tumors, including ACC [8]. This process alters intercellular associations, cell polarity, cytoskeletal structure, and cell-matrix adhesion, potentially promoting cell movement and invasion [9]. Consequently, new drugs aimed at CSCs could offer alternative approaches to enhance patient survival rates.

High-throughput screening (HTS), a large-scale method for identifying potential compounds from drug libraries, has emerged as a crucial aspect of early-stage drug discovery [10,11]. Despite ACC’s rarity, diverse morphology, and varying biological behavior, advances in developing novel therapeutic strategies have been hindered. Therefore, our study aims to bridge the identified gap for effective ACC management drugs by conducting HTS of epigenetic drugs capable of halting tumor invasion by preventing EMT and targeting CSCs.

## 2. Results

**Histone modification library and its targets.** Histone modification has emerged as a promising avenue in cancer therapy. Histones, proteins that play a crucial role in packaging DNA into chromatin, can influence gene expression patterns [12,13]. Aberrant histone modifications have been linked to various diseases, including cancer, where they can lead to dysregulated gene expression and promote tumorigenesis [14,15]. In this study, we investigated the cytotoxic potential of 157 different histone modification compounds on adenoid cystic carcinoma cells. This library targets several epigenetic mechanisms, including histone methyltransferases, bromodomains, histone deacetylases, and histone demethylases. The most effective compounds, capable of inducing tumor cell death, targeted histones methyltransferases and deacetylases, along with Sirtuin inhibitors, histone demethylases, and acetyltransferases (Figure 1A). Interestingly, many compounds were observed to induce modifications to the nuclear size of tumor cells, suggesting a relaxation or ‘decondensation’ of the chromatin structure (Figure 1B). Indeed, such components induced the acetylation of histone H3 at lysine 9 (H3K9ac)—a crucial epigenetic modification associated with transcriptional activation through chromatin relaxation (Figure 1C) [16,17].

**Histone modification-induced activation of EMT and Snail expression.** *Snail*, a transcription factor, plays a central role in the invasion and metastasis of salivary gland tumors [18]. *Snail’s* primary role is in inducing epithelial–mesenchymal transition (EMT), a biological process where epithelial cells acquire mesenchymal cell phenotypes, which enhance their migratory and invasive capabilities [19,20], and has been related to poor prognosis in several salivary gland carcinomas. Therefore, the identification of novel small molecules capable of interfering with Snail’s function in ACC can provide new adjuvant therapies to manage tumor invasion and metastasis.

Initially, we sought to explore the extent to which histone modifications could affect the morphology of ACC tumor cells. Interestingly, we found that 38.2% of the histone modification compounds activated an EMT phenotype in ACC cancer cells. In comparison, 55.4% did not induce EMT, and 6.36% of the compounds induced a strong cell death response (Figure 2A,B). In a manner analogous to the activation of the EMT phenotype, our findings identified a total of 26 compounds, representing 16.5% of our sample, which successfully prompted the accumulation of Snail in ACC cells. These compounds mainly targeted HDACs, bromodomains, histone methyltransferases, and Sirtuins (Figure 2C).

**Identification of histone modification compounds capable of inducing histone acetylation and reducing the protein expression of Snail.** Histone acetylation is a key epigenetic modification that influences gene expression and has been identified as a crucial mechanism for sensitizing tumor cells to therapy [21]. In this process, acetyl groups are added to the histone proteins, causing chromatin relaxation and promoting gene transcription [22]. This alteration primes tumor cells, including ACC tumors, to be more responsive to therapy by promoting the expression of tumor suppressor genes and other anticancer genes and enhancing DNA accessibility to therapeutic drugs and radiation [3,23,24,25].

In our study, we sought to identify novel histone modifiers presenting a dual function of inducing histone acetylation and, therefore, sensitizing tumor cells to chemotherapy and reducing Snail levels from ACC cells. From the initial compound screened, we identified RG2833 (CAS# 1215493-56-3), CI994 (CAS# 112522-64-2), LMK 235 (CAS# 1418033-25-6), M344 (CAS# 251456-60-7), KD5170 (CAS# 940943-37-3), Scriptaid (CAS# 287383-59-9), and Entinostat (CAS# 209783-80-2) as having the capability to reduce Snail levels, while inducing histone acetylation (Figure 3A). Representative immunofluorescence of ACC tumor cells stained for Snail and H3K9 along computer segmentation module used to identify and quantify positive cells in a high throughput approach (Figure 3B).

**Validation of selected histone modifiers on ACC biology.** In order to validate our drug screening results, we decided to evaluate our top compounds in a series of assays, including cellular migration, cell death, cancer stem cell viability and sphere size, and Snail and H3K9 expression levels. From the original seven hits identified in our drug screening, we have pursued six compounds and decided to leave aside Entinostat as we have previously published its effects on PDX models of ACC [24]. Initially, we assessed the effects of the selected compounds on the migratory potential of ACC cells (Figure 4A). Our assay was performed for over 30 h, and data was collected every 6 h. Scriptaid was the most powerful compound capable of significantly disrupting cellular migration starting after 12 h of seeding cells (Figure 4B, **** *p* < 0.0001). CI994, an HDAC inhibitor, also disrupted cellular migration after 24 h of initiating the assay (Figure 4B, T24 ** *p* < 0.01).

We also observed that four out of the six compounds analyzed could induce low but statistically significant levels of cell death ranging from 4.8% to 6.3% of total ACC cells cultured in monolayer compared with baseline cell death levels found in control cells (0.7%) (**** *p* < 0.0001) (Figure 4C).

We further decided to explore the effects of our top histone modifier candidates on the population of cancer stem cells. Initially, we identified the population of CSCs from our ACC cell line by detecting high levels of aldehyde dehydrogenase (ALDH) commonly found in normal and cancer stem and progenitor cells of various lineages [26,27]. We observed that ACC cells receiving the histone modifier compounds LMK 235, M344, and Scriptaid had a significant reduction in the number of CSCs (* *p* < 0.05, ** *p* < 0.01, and *** *p* < 0.001) (Figure 5A). Complementary to the identification of aldehyde dehydrogenase, we also cultured and analyzed tumorspheres as a functional assay for CSCs (Figure 5B). We found that tumorspheres present an increased number of dead cells as judged by the accumulation of propidium iodide when exposed to Scriptaid (44% of cells), M344, and LMK 235 (both with 34% of cells) compared with vehicle control (18% of cells) and positive control (H_2_O_2_—54% of cells) (Figure 5C).

**Targeted modulation of Snail and H3K9ac.** *Snail* gene silencing has emerged as a potential therapeutic intervention in the context of cancer treatment [28]. This gene plays a pivotal role in epithelial–to–mesenchymal transition (EMT), a crucial process for the progression and metastasis of cancer. High expression of the Snail gene is often noted in progressive cancer, leading to higher invasive capability and survival of cancer cells [29]. Silencing the *Snail* gene can impair the cell migration and invasion process, thus potentially restricting cancer metastasis [28,30]. The identification of novel Snail inhibitors capable of attenuating the pathological EMT process and eventually restricting cancer progression and metastasis has attracted extensive attention in recent years [31]. Like Snail inhibitors, histone modification agents are emerging as feasible therapeutic strategies capable of relaxing the DNA, making it more accessible for transcription and ultimately leading to changes in gene expression while sensitizing tumor cells to chemotherapy [25]. Here, we explored the potential effects of our top compounds on the protein expression levels of Snail and acetylation of histone H3K9 using flow cytometry. Interestingly, we observed that the majority of our compounds were able to suppress the expression levels of Snail in vitro (Figure 5D). KD5170 was the most powerful inhibitor of Snail along M344 (**** *p* < 0.0001) in our validation assay, followed by LMK 235 and Scriptaid (** *p* < 0.01, * *p* < 0.05). We also found that all compounds could induce histone acetylation (H3K9) beyond the vehicle group, except for M344 (Figure 5E).

**The overall impact of selected histone modification agents on ACC tumor cells.** We have used a panel of histone modifier compounds organized by their targets. Our compounds targeted from HDACs to Sirtuins, histone demethylases, and bromodomains to cite a few (see Figure 1 for the complete target list). Interestingly, our top candidates capable of reducing the protein levels of Snail while inducing histone acetylation were found to mainly target HDACs. In order to rationalize our results and present the small molecule capable of achieving the best results, we have developed a hierarchy graphic to distribute the top inhibitors among their ability to inhibit tumor cell migration, induce ACC cell death, reduce CSCs, induce cell death in tumorspheres, induce the acetylation of histone H3, and reduce the protein levels of Snail (Figure 6). We found that Scriptaid was the only small molecule capable of providing positive results in all six validation assays. Following Scriptaid, LMK235 showed positive results in five of the assays, M344 in four, CI994 and KD5170 in three assays, and RG2833 was able to induce cellular death of tumorspheres and histone acetylation of tumor cells.

## 3. Discussion

Current research on ACC of the salivary glands investigates various signaling pathways to improve therapeutic strategies. Notably, the Notch, Hedgehog, and MYB-NFIB signaling pathways are of primary interest. The MYB-NFIB fusion gene, in particular, is a prominent feature of ACC and has been linked to oncogenic transcriptional activity. Additionally, the protein kinase C (PKC) pathway, the fibroblast growth factor receptor (FGFR) pathway, and the androgen receptor (AR) pathway are also being studied for their roles in ACC. Recent studies are investigating the potential of epigenetic drugs to either enhance the responsiveness or directly treat various forms of malignancies. Leveraging a drug library of histone modifiers in cancer research offers numerous advantages. Histone modifications play critical roles in regulating gene expressions; thus, drugs that can modify these events hold immense potential in altering disease progression. These libraries comprise a broad array of bioactive compounds targeting crucial histone-modulating enzymes, such as demethylases, acetyltransferases, and deacetylases, to influence carcinogenic pathways. Hence, screening histone-modifying drugs provides an efficient route to identify promising candidates with potential anticancer properties, bringing new therapeutic avenues for cancer management and paving the way for personalized medicine.

Adenoid cystic carcinoma (ACC) from the salivary gland is a malignant neoplasm known for its slow growth and indolent clinical course. Despite this, the long-term prognosis remains poor, with a survival rate of patients receiving surgical resection of approximately 35% after 20 years [32]. Therapeutic options for ACC are limited due to its heterogeneous nature and the scarcity of available samples for research. Consequently, effective systemic chemotherapy protocols are still lacking, necessitating exploring new treatment approaches.

Our study aimed to identify potential candidate drugs for treating ACC by conducting a sophisticated phenotypic screening combined with automated image data analysis through high-throughput screening (HTS). This technique allowed us to efficiently screen a large chemical epigenetic library and identify promising compounds. We also chose to use the UM-HACC-2A cell line in this study as it contains the classical MYB-NFIB fusion protein found in 60–80% of salivary ACC patients and high expression of proteins typical of ACC like c-Myb and p63, thereby well representing the biology and genetics of the predominant cases of ACC tumors in patients [33]). Also noteworthy is that UM-HACC-2A cells maintain the capacity to form tumors in mice when introduced directly into their submandibular glands. Not only do these tumors thrive, but they also successfully reproduce the cribriform structure of their original tumors [33].

Epithelial–mesenchymal transition (EMT) plays a crucial role in tumor invasion and cell migration in various neoplasms. The β-catenin/TCF/LEF complex is known to stimulate the activation of EMT by inducing the expression of Snail1, a zinc-finger transcription factor implicated in tumor genesis and metastasis, providing a key mechanism for cell plasticity, migration, and invasion [34,35,36]. We used morphological analysis and Snail marker expression to confirm EMT activation and identify drugs that could specifically act on ACC cell invasion. Histone deacetylase inhibitors (HDACi) caused rapid changes in ACC cell morphology and increased Snail staining. At the same time, other epigenetic drugs inhibited Snail expression, suggesting their potential to reduce tumor cell invasion.

Epigenetic modifications are critical in tumor development and progression, affecting gene expression and cell behavior [37]. Chromatin acetylation through HDAC inhibition has shown promising results in acting on cancer stem cells (CSCs) [8,38,39,40]. In our study, several hit compounds inhibited Snail expression and hyperacetylated H3K9, suggesting their potential to disrupt CSCs.

During our drug screening of histone modification compounds, we identified seven hit compounds, including Scriptaid, LMK235, M344, KD5170, RG2833, CI994, and Entinostat. We have previously explored the effects of Entinostat in ACC tumors; therefore, we chose to exclude this compound from the current analysis [24]. We found that Scriptaid and CI994 significantly inhibited tumor migration, while Scriptaid, KD5170, M344, and LMK235 inhibited Snail expression in ACC cells. Notably, Scriptaid, M344, and LMK 235 effectively disrupted the number of tumor cells expressing high levels of aldehyde dehydrogenase, a marker for CSCs.

Our findings suggest that Scriptaid may be a promising drug for managing ACC due to its ability to induce cancer cell death, reduce cancer migration, reduce ALDH+ cancer cells, induce the acetylation of cancer cells, and reduce the levels of the EMT marker Snail. Previous studies have also demonstrated the antitumor activity of Scriptaid in other aggressive cancer phenotypes, highlighting its potential as an efficient HDAC inhibitor with lower toxicity.

It is important to acknowledge that our findings were obtained from single-agent administration, and combined therapy may yield different results, especially in overcoming CSC chemoresistance. Moreover, Scriptaid’s high stability and limited toxicity in normal cells make it a potential therapeutic agent against various cancer types, either as a standalone drug or in combination therapy.

## 4. Materials and Methods

### 4.1. Cell Line and Culture Conditions

We utilized the UM-HACC-2A cell line, derived from a T3N1M0 adenoid cystic carcinoma (ACC) from a minor salivary gland at the base of the tongue presenting the classical MYB-NFIB translocation, as described by Warner et al. (2018) [41]. The cells were cultured in DMEM (Hyclone) supplemented with sodium pyruvate, 10% fetal bovine serum (Hyclone), 200 mM L-glutamine (Gibco, Waltham, MA USA), 20 ng/mL human epidermal growth factor (Sigma-Aldrich, St. Louis, MO, USA), 400 ng/mL hydrocortisone (Sigma-Aldrich), 5 µg/mL insulin (Sigma-Aldrich), and a cocktail of antibiotic/antimycotic solution containing Contains 10,000 units of penicillin, 10,000 μg of streptomycin, and 25 μg of Amphotericin B per mL (HyClone, Logan, UT, USA) in a 10 cm culture dishes. Cells were maintained in a 5% CO_2_ humidified incubator at 37 °C and subcultured when reaching 70% confluency.

### 4.2. Histone Modifiers Library and High-Throughput Screening (HTS)

We seeded ten thousand ACC cells in two 96-well plates (Cellvis, Mountain View, CA, USA) using a robotic liquid handler system (Opentrons OT-2, Long Island City, NY, USA) and incubated them for 24 h in a 5% CO_2_ humidified incubator at 37 °C. After 24 h, we treated the cells with 10 µL of each compound from a histone modification library for HTS (DiscoveryProbe™ Histone Modification Library—APExBIO, Boston, MA, USA). The library comprises 157 histone modification-related compounds divided into 2 96-well plates at an initial concentration of 10 mM diluted in DMSO. We analyzed cell viability and morphology changes 24 h after drug administration using a high-content imaging system (Molecular Devices ImageXpress Micro 4, San Jose, CA, USA).

### 4.3. Immunofluorescence

For immunofluorescence, we fixed the cells with 4% paraformaldehyde for 15 min at room temperature, followed by blocking for nonspecific binding antigen using 1% PBS/BSA, and then incubated overnight with anti-mouse Snail (Cell Signaling Technology clone L70G2, CST: Danvers, MA, USA) or anti-rabbit H3K9 (Cell Signaling Technology clone C5B11) antibodies. Subsequently, cells were washed and incubated with anti-mouse Alexa-Fluor 488 or anti-rabbit Alexa-Fluor 647 conjugated secondary antibodies (Thermo Fisher, Waltham, MA, USA). DNA content was stained with Hoechst 33,342 (Invitrogen) for nuclear visualization and propidium iodide (PI—Invitrogen, Paisley, UK) to assess cell death. Images were captured using the ImageXpress Micro 4 system.

### 4.4. Flow Cytometry

UM-ACC-2A cells were seeded in 6-well plates at a density of 1 × 10^6^ cells/plate using salivary gland culture medium (SGM) for 24 h before treatment. Drugs were added to the medium and after 24 h of treatment, cells were collected, fixed in 70% ethanol, and stained with anti-Snail and anti-H3K9 antibodies following analysis using a BD Accuri C6 Plus flow cytometer. Each experimental group was performed in sextuplicate, and the flow cytometer was set to read 10,000 events in each replicate [42].

### 4.5. Scratch Assay

Cells were seeded at a density of 2 × 10^5^ on 24-well culture plates in triplicate for each epigenetic drug identified. After 24 h of culture, each well containing cells at 100% confluency were scraped to create cell-free linear wound areas. After three washes with PBS, cells were treated with each drug (10 µM), and the wound closure was evaluated at 0, 6, 12, 24, and 30 h [38,43]. Images of the wound areas were collected at each time point using an inverted microscope (EVOS Cell Imaging Systems, Seoul, Republic of Korea). The percentage of the relative wound closure was measured using image analysis software (ImageJ 1.52v, Bethesda, MD, USA).

### 4.6. Tumorsphere Assay

To evaluate the ability of tumor cells to grow in suspension and form spheres, we cultured UM-HACC-2A cells (5 × 10^3^) in ultra-low attachment round-bottom plates (Corning^®^ Elplasia^®^ 96-well Black/Clear Round Bottom Ultra-Low Attachment—Thermo Fisher Scientific, Waltham, MA, USA) [44,45,46]. Cells were incubated for 3 days to allow spheres to form. Following this, we added the histone modifier drugs identified in the HTS to the plates at a final concentration of 10µM and closely monitored the spheres for 48 h. Images of the spheres were obtained and quantified using Molecular Devices ImageXpress Micro 4.

### 4.7. Statistical Analysis

We performed statistical analysis using GraphPad Prism (Prism 10 for macOS) and data visualization using Flourish (Canvas, London, UK, https://app.flourish.studio). For the drug toxicity index assay, we used one-way ANOVA. The Snail flow assay was assessed using one-way ANOVA followed by Dunnett’s correction test. The cell migration assay (scratch assay) was analyzed using two-way ANOVA. Statistical significance was denoted by asterisks (* *p* < 0.05; ** *p* < 0.01; *** *p* < 0.001, **** *p* < 0.0001; NS; *p* > 0.05).

## 5. Conclusions

In conclusion, our study has successfully utilized high-throughput screening to identify potential small molecule modulators of epithelial–mesenchymal transition (EMT) in adenoid cystic carcinoma (ACC). The data obtained from screening 157 drugs present exciting opportunities for further investigation and drug discovery in ACC treatment.

Notably, Scriptaid emerged as a standout candidate, exhibiting multiple favorable properties. It displayed promising levels of cellular toxicity, effectively inhibited Snail expression and flow, induced histone H3K9 hyperacetylation, and effectively reduced ACC cell migration. Moreover, Scriptaid demonstrated the ability to disrupt ACC cancer stem cells (CSCs), presenting a potential avenue for addressing the challenges these cells pose in cancer progression.

The findings regarding Scriptaid’s multifaceted effects make it a strong contender for future preclinical studies in the treatment of salivary gland ACC. Its potential to address the limitations of current treatment options offers hope for improved outcomes and patient management.

Our study has opened up a promising path for exploring Scriptaid as a potential therapeutic strategy for salivary gland ACC. However, further research and preclinical studies are necessary to fully validate its efficacy and safety. The discovery of novel drugs presenting the ability to target EMT and CSCs in ACC holds significant potential to transform the treatment landscape and improve patient outcomes. As such, Scriptaid warrants attention and investment in future investigations to unlock its full therapeutic potential in ACC and possibly other malignancies.

## Figures and Tables

**Figure 1 ijms-25-01646-f001:**
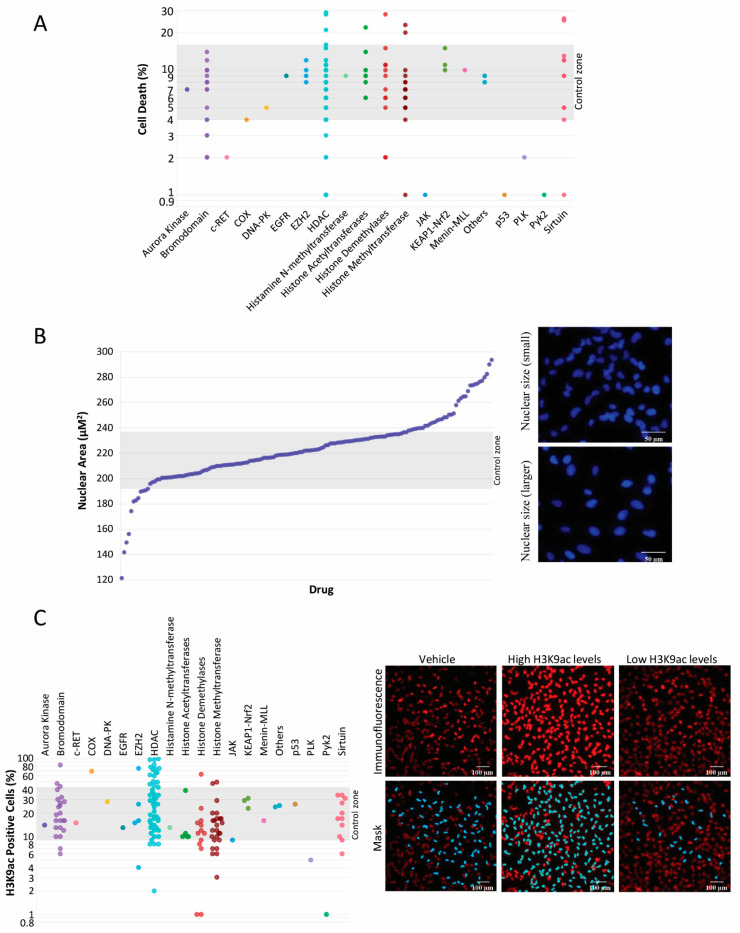
Panel (**A**) represents a library of 157 histone modifiers, distributed relative to their respective drug targets (X-axis), which displays the individual capability of each drug to initiate cell death in UM-HACC-2A cell line as measured by propidium iodide staining (Y-axis). Panel (**B**) depicts a scatter plot of various compounds sorted by their potency to induce alterations in nuclear size, serving as a proxy for chromatin reorganization. The data reflected in this plot were collected via Hoechst 33342 immunofluorescence staining of ACC cells. Scale bar 50 µm. Panel (**C**) illustrates a scatter diagram wherein histone modifiers are plotted according to their specific drug target and efficacy in promoting the acetylation of histone H3K9. The data represent initial high-throughput drug screening containing one read for each of the 157 histone modification compounds after 24 h of treatment. Scale bar 100 µm.

**Figure 2 ijms-25-01646-f002:**
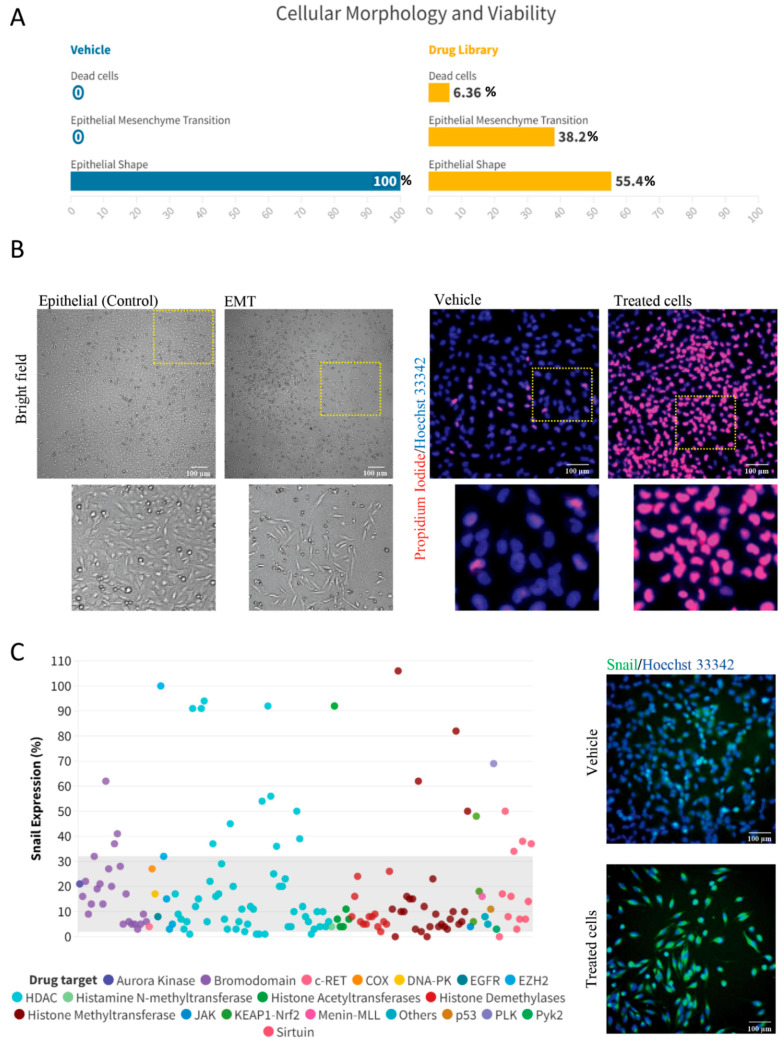
Demonstration of EMT activation and discovery of Snail inhibitors. (**A**) This panel displays a line chart reflecting the proportion of ACC cells exhibiting the EMT phenotype in the Vehicle group and the Drug library. While 55.4% of the library compounds do not stimulate EMT, 38.2% can induce varying degrees of EMT in ACC cells. (**B**) This includes representative brightfield images showcasing ACC cells exhibiting the EMT phenotype post-compound administration and immunofluorescence images delineating cell death, as determined by propidium iodide uptake (red). (**C**) This panel features a scatter plot of 157 histone modification compounds. Each compound is color-coded according to its specific drug target and plotted depending upon its efficiency to enhance or inhibit Snail protein levels in ACC cells. Accompanying are the immunofluorescence images of ACC cells stained with Hoechst 33342 and anti-Snail antibody. Scale bar 100 µm. Data from initial high-throughput drug screening containing an average of 9 ROI for each well of PI, Hoechst 33342 and Snail for each of the 157 histone modification compounds (1 well/drug) after 24 h of treatment.

**Figure 3 ijms-25-01646-f003:**
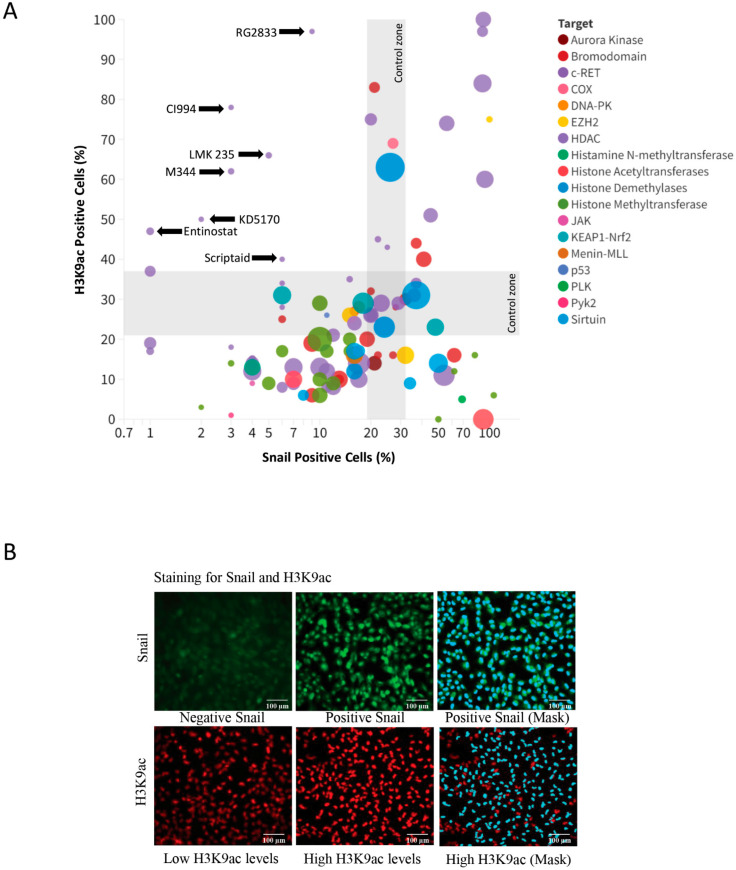
Identification of compounds inducing histone acetylation and suppressing Snail protein levels during drug screening analysis. (**A**) Scatter plot of histone modifiers featuring seven promising candidates that facilitate increased histone acetylation and simultaneously demonstrate reduced Snail protein expression in ACC cells. (**B**) Representative images were obtained from the immunofluorescence staining of Snail and histone H3K9, which were used to pinpoint the top candidates. The accompanying image exhibits an example of automated image segmentation, delineated in blue. Data from initial high-throughput drug screening containing average of 9 ROI for each well of H3K9, Hoechst 33342 and Snail for each of the 157 histone modification compounds (1 well/drug) after 24 h of treatment. Scale bar 100 µm.

**Figure 4 ijms-25-01646-f004:**
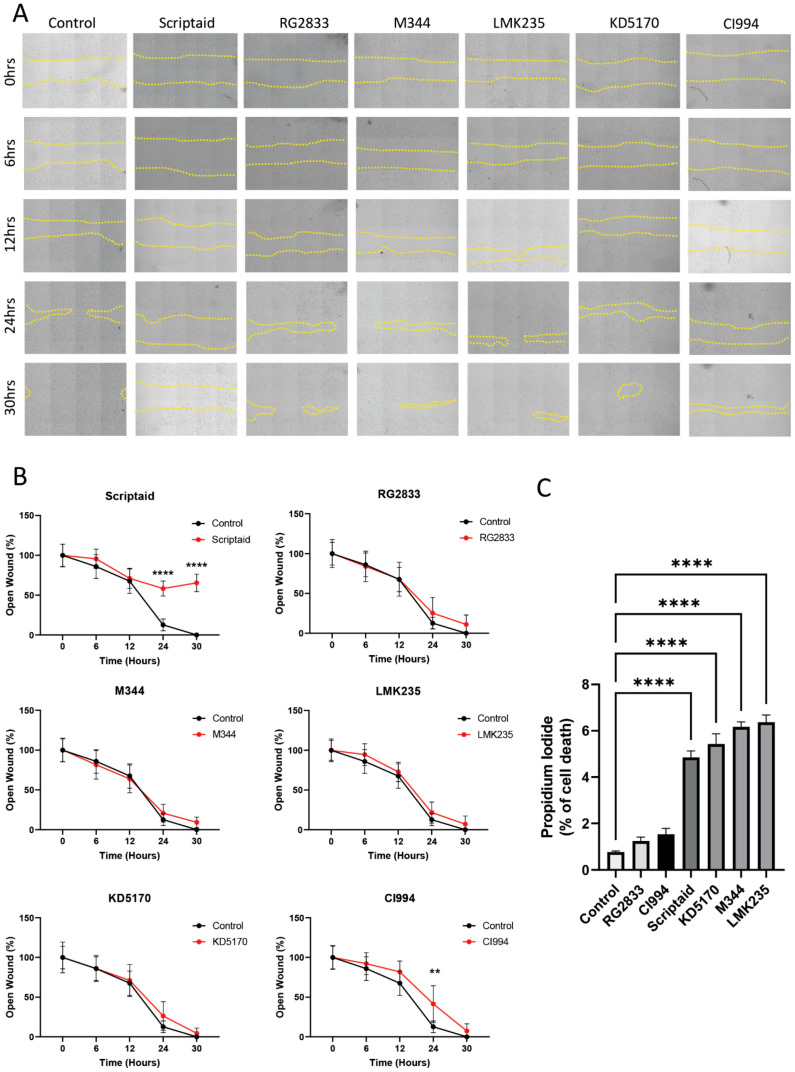
Assessment of cellular migration. (**A**) Illustrated here is a scratch assay conducted on UM-HACC-2A cell line administered with six drug compounds and monitored over 30 h. Yellow dashed lines indicate open wound area (4× magnification). (**B**) This panel quantifies the total open area corresponding to each compound. Notably, only Scriptaid and CI994 display the capacity to reduce ACC migration (** *p* < 0.01, **** *p* < 0.0001). (**C**) The bar graph signifies the effects of each of the six candidate compounds on ACC cell death (**** *p* < 0.0001). We conducted each drug migration assay in triplicate, and each replicate had four measurements.

**Figure 5 ijms-25-01646-f005:**
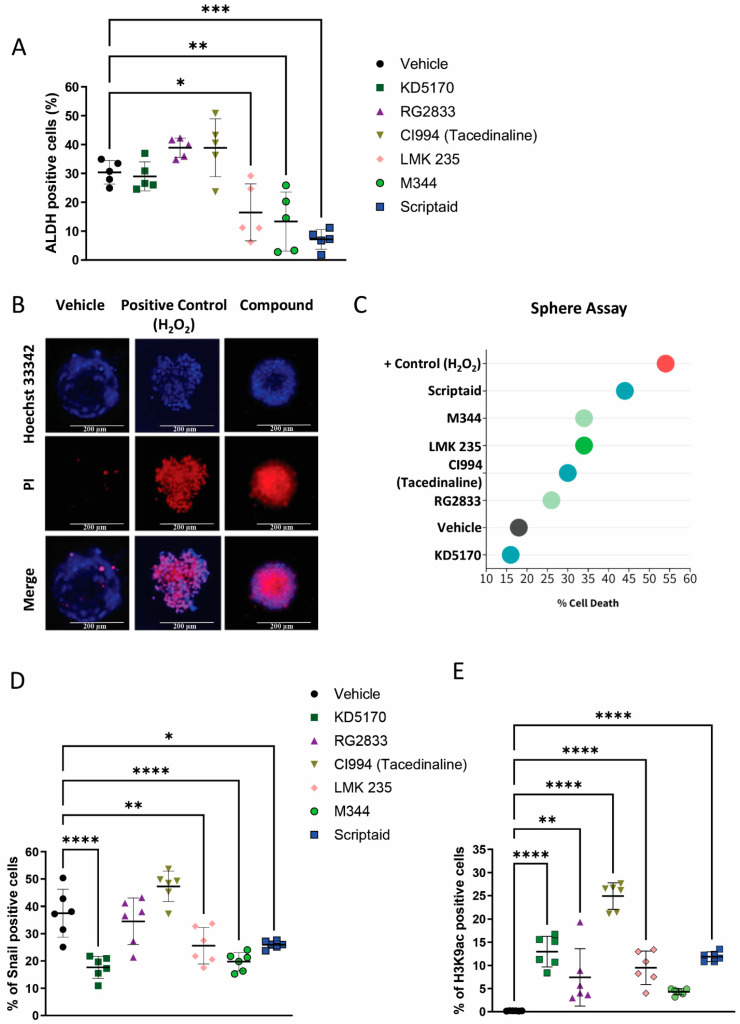
(**A**) Graphical representation of the impacts of the top six compounds on the population of cancer stem cells (CSCs) derived from ACC cells determined via the measurement of aldehyde dehydrogenase (ALDH) enzymatic activity (* *p* < 0.05, ** *p* < 0.01, *** *p* < 0.001). We conducted each flow cytometry assay in quintuplicate per drug. (**B**) A representative immunofluorescence image highlights cancer spheres stained with Hoechst 33342 and propidium iodide (PI). The positive control group was treated with H_2_O_2_, while baseline levels of cell death were noted in tumorspheres cultured in DMEM supplemented with 10% FBS. (**C**) Scatter plot representing tumorspheres distributed according to the percentage of cell death, consistent with the impacts of diverse compounds. (**D**) Flow cytometry data illustrates the percentage of Snail-positive cells following the administration of all six identified compounds. It is important to note that four out of the six compounds successfully reduced Snail levels (* *p* < 0.05, ** *p* < 0.01, **** *p* < 0.0001). We conducted each flow cytometry assay in sextuplicate per drug. (**E**) The data present cells that tested positive according to the flow cytometry of histone H3K9ac. Except for the M344 compound, all were capable of inducing tumor cell acetylation above the vehicle group. We conducted each flow cytometry assay in sextuplicate per drug. (** *p* < 0.01, **** *p* < 0.0001). Scale bar 200 µm.

**Figure 6 ijms-25-01646-f006:**
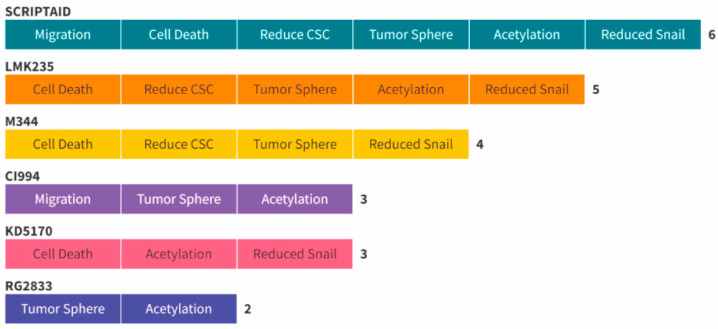
This panel showcases a hierarchical graphic delineating the overall distribution of all six evaluated compounds, arranged according to their efficacy in disrupting distinct biological hallmarks of ACC cancer cells.

## Data Availability

Data are available upon request to the corresponding author.

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
