# Peer review of "Novel Epigenetic Modifiers of Histones Presenting Potent Inhibitory Effects on Adenoid Cystic Carcinoma Stemness and Invasive Properties"

_ijms, 2024, doi:10.3390/ijms25031646_

Round 1

Reviewer 1 Report

Comments and Suggestions for Authors

ACC is a rare and slow-growing cancer known for its perineural invasion and delayed distant metastasis. In this current piece of work, the authors utilized high-throughput screening of epigenetic drugs, and identified Scriptaid and LMK235 as promising candidates for treating ACC invasion by disrupting tumor and cancer stem cell properties, potentially improving patient outcomes and inspiring further research in this challenging area. However, certain aspects need addressing before the study can be published.

Major points:

1. One of the major concerns is the use of only one cell line, UM-HACC-2A, which complicates understanding the reproducibility of the study. It is suggested to include experiments with at least one additional Adenoid Cystic Carcinoma (ACC) cell line to validate that observed phenomena are not cell-dependent.

2. Additionally, when discussing the activation of Epithelial-Mesenchymal Transition (EMT), the authors should assess EMT markers (E-cadherin, N-cadherin, Vimentin, etc.) and their protein or genomic level expression.

Minor points:

1. Lack of major references in the MS, a suggestion to include references for methods and materials mentioned.

2. Clarification on the number of times each experiment was conducted (either in the methods or figure legends).

3. Scale bar is missing in microscopy images.

Comments on the Quality of English Language

Moderate level of English improvement is required.

Reviewer 2 Report

Comments and Suggestions for Authors

This is an interesting manuscript about new candidates for adenoid tumors.

In the Background, it is important to describe the drugs already use and the efficacy of them in this cancer. 

It is important to describe why the authors choose this kind of tumor to study.

Besides histones, another mechanisms have been already explored in this cancer?

In Methods Section

Include which antibiotic was used in the culture essay

Include how many events were used in cytometry essay

In results

Please replace Figs 1-5. These figures are with poor quality.

Sincerely,

Comments on the Quality of English Language

Some minor errors need to be improved
